# A fluorescence viewer for rapid molecular assay readout in space and low-resource terrestrial environments

Kristoff Misquitta[1], Bess M. Miller[2], Kathryn Malecek[3], Emily Gleason[4], Kathryn Martin [4], Chad M. Walesky[2¤], Kevin Foley[5], D. Scott Copeland[5], Ezequiel Alvarez Saavedra[4], Sebastian Kraves [4]*

1 Stuyvesant High School, New York, NY, United States of America, 2 Division of Genetics, Brigham and Women's Hospital, Harvard Medical School, Boston, MA, United States of America, 3 Massachusetts Institute of Technology, Cambridge, MA, United States of America, 4 miniPCR bio, Cambridge, MA, United States of America, 5 Boeing Defense, Space & Security, Berkeley, MO, United States of America

¤ Current address: PhenomeX, Emeryville, CA, United States of America
* seb@minipcr.com

**Data Availability Statement:** All relevant data are within the paper and its Supporting Information files.

## Abstract

Fluorescence-based assays provide sensitive and adaptable methods for point of care testing, environmental monitoring, studies of protein abundance and activity, and a wide variety of additional applications. Currently, their utility in remote and low-resource environments is limited by the need for technically complicated or expensive instruments to read out fluorescence signal. Here we describe the Genes in Space Fluorescence Viewer (GiS Viewer), a portable, durable viewer for rapid molecular assay readout that can be used to visualize fluorescence in the red and green ranges. The GiS Viewer can be used to visualize any assay run in standard PCR tubes and contains a heating element. Results are visible by eye or can be imaged with a smartphone or tablet for downstream quantification. We demonstrate the capabilities of the GiS Viewer using two case studies–detection of SARS-CoV-2 RNA using RT-LAMP and quantification of drug-induced changes in gene expression via qRT-PCR on Earth and aboard the International Space Station. We show that the GiS Viewer provides a reliable method to visualize fluorescence in space without the need to return samples to Earth and can further be used to assess the results of RT-LAMP and qRT-PCR assays on Earth.

## Introduction

Fluorescence-based molecular assays have emerged as a powerful asset in clinical diagnostics and environmental monitoring. Their sensitivity and scalability–manifest in techniques like quantitative polymerase chain reaction (qPCR), loop-mediated isothermal amplification (LAMP), and enzyme-linked immunosorbent assay (ELISA)– have made them the frequent method of choice for detecting gene expression, protein activity, and various metabolites in samples of interest [1]. However, owing to high cost, complexity, and the requirement to operate within a centralized lab framework, state-of-the-art fluorescence-based assays have faced

**Funding:** The author(s) received no specific funding for this work.

**Competing interests:** The authors have declared that no competing interests exist.

major barriers to adoption in field science and low-resource or disaster settings [2]. Similar hurdles are faced aboard the International Space Station (ISS), where gene expression studies are fundamentally limited by equipment availability and the need to return samples to Earth, forcing vibrational and temperature stresses from atmospheric re-entry to be introduced before data analysis [3]. While Wetlab-2's launch of a qPCR machine to the ISS has enabled real-time fluorescence analyses in orbit, it does not provide solutions for other types of fluorescence-based assays, including protein detection [4].

Portable fluorescence devices have emerged as answers to such limitations. Previous iterations have run the gamut from low cost to high manufacturability, exploiting technologies like 3D printing or smartphone integration for easier data collection [5]. The cost of building some fluorescence sensors falls as low as $15, and sensors have already been used for purposes such as quantifying RNA transcription *in vitro* [6]. There is further interest in reducing the training and calibration needs for use of these devices, as well as integrating fluorescence capabilities into other existing biological workflows, including PCR, which are concurrently benefitting from reductions in price and increases in accessibility. To address these needs, we present an inexpensive, portable, and multipurpose device for visualizing fluorescence in biological workflows, called the Genes in Space Fluorescence Viewer (based on the instrument described in US patent 10,983,058 [7]).

Here we test the performance of the GiS Viewer in two different experimental settings: reading out the results of qRT-PCR assays on the International Space Station and LAMP-based detection of SARS-CoV-2 RNA on Earth. The two methodologies serve as both a rigorous evaluation of the GiS Viewer and, more broadly, a proof-of-concept of the utility of portable fluorescence in the domains of *in situ* biological research and point-of-care molecular testing. In the future, portable sensors like the GiS Viewer may accelerate diagnostics by reducing burdens on centralized labs or facilitating novel approaches to research questions that were previously intractable due to cost or resource limitations.

## Results

The GiS Viewer permits relative red-green fluorescence among samples to be evaluated in real time with the naked eye or recorded with a second device for data analysis and sharing (**Fig 1**). The GiS Viewer consists of a heated block that holds sample tubes, an internal excitation light source, and a viewing window (**Fig 1A**). Through the window on the front of the device an observer can directly visualize experimental samples while assays are ongoing. Alternatively, a camera adapter can be attached to the viewing window to hold a commercial phone or tablet, allowing samples to be imaged (**Fig 1B**). As such, the device is a suitable endpoint for many protein and nucleic acid-based workflows, including the assessment of gene expression and protein abundance or activity.

To demonstrate the utility of the GiS Viewer for diagnostic applications such as viral testing, we first set out to detect increasingly low amounts of SARS-CoV-2 RNA using the reverse transcription loop-mediated isothermal amplification (RT-LAMP) method. This method was chosen in lieu of endpoint PCR because reaction progress can be monitored continuously in the GiS Viewer using either the naked eye or by movie or picture recording using a mobile phone, tablet, or other recording device. Continuous monitoring allows for visualization of the increase in fluorescence over the course of the reaction. We prepared duplicate RT-LAMP reactions containing 1000, 500, and 100 copies of synthetic SARS-CoV-2 RNA and incubated them at a constant temperature of 65°C for 30 minutes. The set of primers, including a molecular beacon, targeted the N2 region of the virus [8]. 1000, 500, and 100 copies of the viral RNA could be reliably detected in the viewer (**Fig 2**).

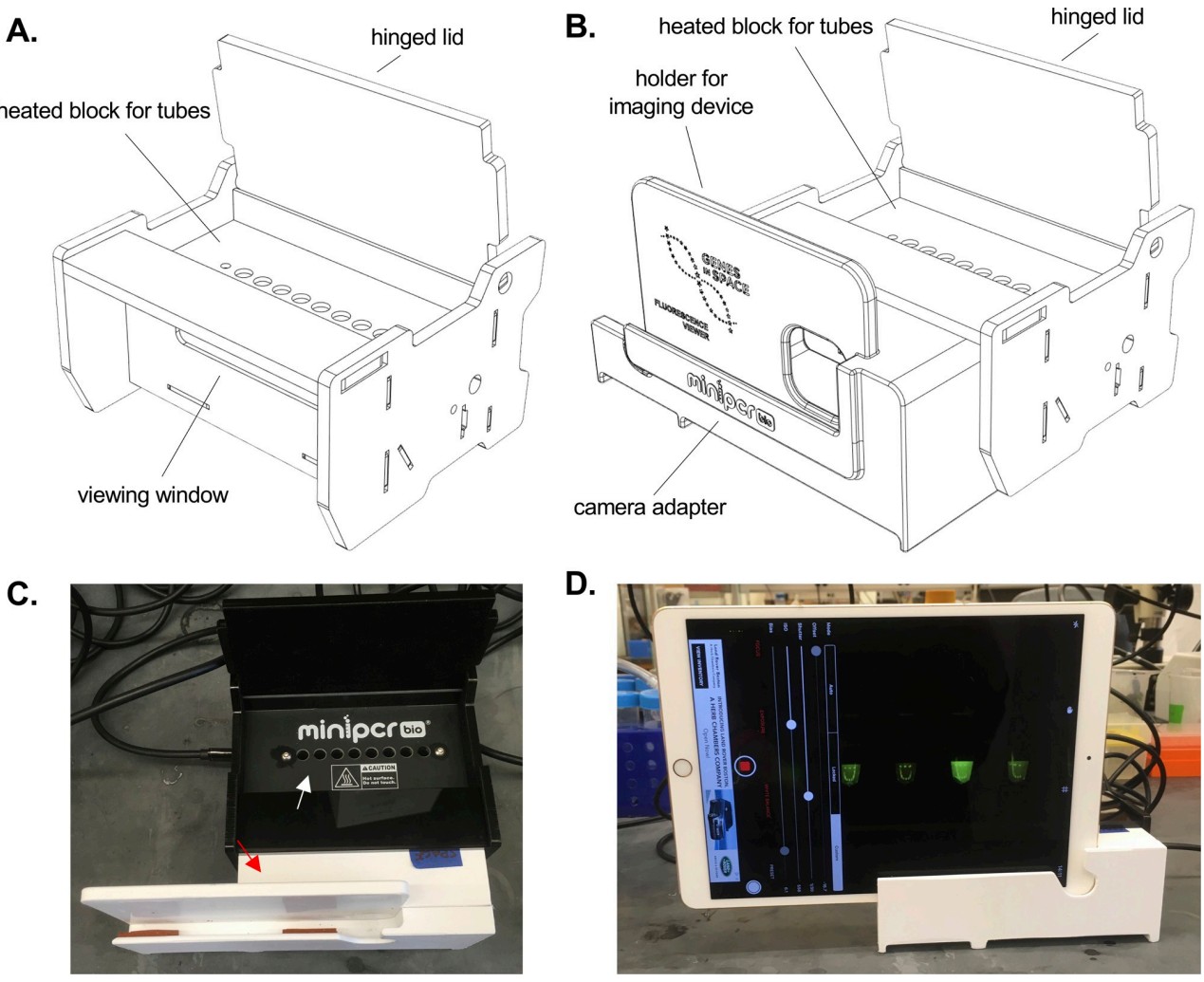

**Fig 1. Genes in Space Viewer. A-B.** 3D schematics of the GiS Viewer alone (**A**) or with attached camera adapter (**B**). The GiS Viewer contains a heated block designed for holding 0.2 mL tubes. A viewing window on the front of the device allows direct visualization of samples while the experiment progresses. A camera adapter can also be attached to the GiS Viewer to hold a tablet or phone, allowing images and video to be taken while the experiment is in progress. The GiS Viewer has a hinged lid, which is closed during sample incubation and image acquisition. **C.** Top view of the GiS Viewer with lid open to show inner chamber where samples can be inserted (white arrow). A holder to stabilize a tablet or phone can be inserted in front of the viewing window for imaging (red arrow). **D.** Front image of GiS Viewer with iPad inserted into front holder and imaging app open for fluorescence visualization.

We next asked whether the GiS Viewer could be used in combination with the miniaturized miniPCR® thermal cycler [9–12] to provide a portable, durable method of evaluating gene expression in the field or low resource settings without the need to return samples to a central laboratory. To determine the limit of detection for assessing gene expression in this manner, we first amplified the *18S rRNA* housekeeping gene from a 2-fold dilution series of mouse liver cDNA. RT-PCR reactions were run in the miniPCR and imaged every two cycles between cycles 10–35 in the GiS Viewer. This demonstrated that we are able to detect 3–4 fold differences in gene expression using this technique (**Fig 3A and 3D**).

To ask whether the GiS Viewer could detect perturbation-induced changes in gene expression, we developed a RT-PCR assay to examine the effect of acetaminophen (APAP) on gene expression in the liver. We used a murine liver single cell RNA-sequencing dataset previously

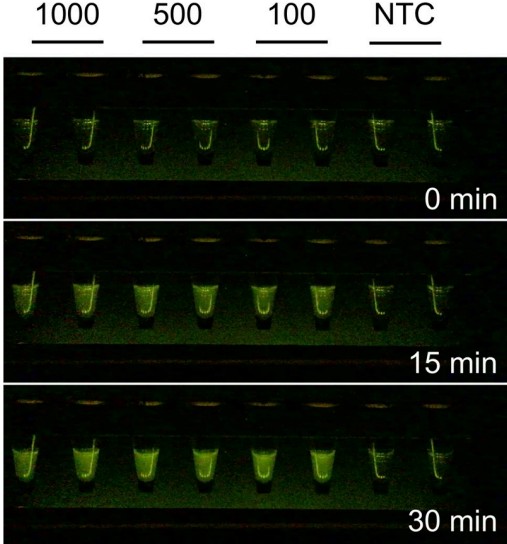

**Fig 2. RT-LAMP detection of SARS-CoV-2 RNA.** Samples containing 1000, 500, or 100 copies of SARS-CoV-2 RNA were used as template for RT-LAMP reactions. Reactions were monitored continuously in the GiS Viewer and maximum signal was observed at 30 minutes. Samples 1–2: 1000 copies of RNA, samples 3–4: 500 copies of RNA, samples 5–6: 100 copies RNA, samples 7–8: no template control. Limit of detection for this assay was 100 copies within 15 minutes.

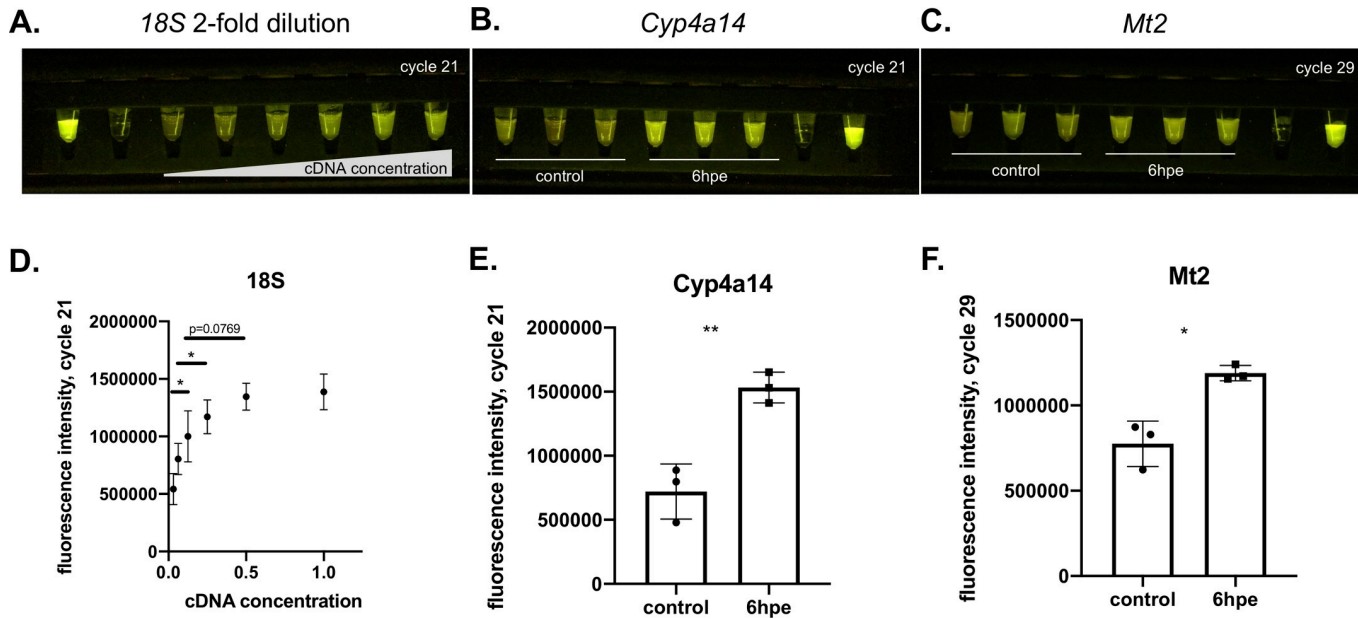

**Fig 3. Rapid readout of gene expression studies using the GiS Viewer. A.** *18S rRNA* RT-PCR reaction with 2-fold dilution series of input liver cDNA run in miniPCR® and imaged in GiS Viewer, shown at cycle 21. Left to right: 10 µM fluorescein, water, cDNA 2-fold dilution series ranging from 0.03125 µl per reaction to 1 µl per reaction. **B.** *Cyp4a14* RT-PCR run in the miniPCR and imaged in GiS Viewer shown at cycle 21, samples 1–3: control liver cDNA from three separate mice, samples 4–6: 6hpe APAP-treated liver cDNA from three separate mice, sample 5: water control, sample 6: 10 µM fluorescein. **C.** *Mt2* RT-PCR run in miniPCR and imaged in GiS Viewer shown at cycle 29. samples 1–3: control liver cDNA from three separate mice, samples 4–6: 6hpe APAP-treated liver cDNA from three separate mice, sample 5: water control, sample 6: 10 µM fluorescein. **D.** Quantification of fluorescence intensity in **A.**, n = 3 per sample. We were able to distinguish a 3–4 fold difference in gene expression and up to a 16-fold dilution of input cDNA. **E.** Quantification of fluorescence intensity in **B. F.** Quantification of fluorescence intensity in **C.** APAP: acetaminophen. hpe: hours post-exposure. Error bars are mean + SD. *p<0.05, **p<0.01, ***p<0.001, ****p<0.0001. Blank and positive control are not included in quantification of fluorescence intensity.

published by one of our labs [13] to identify genes that are significantly up- or downregulated in hepatocytes following acetaminophen treatment. From this dataset, we selected two candidates for further follow up as they demonstrated >1 log fold change in expression between hepatocytes from control mice and those isolated from mice 6 hours post-APAP exposure (hpe): *Cytochrome P450 omega-hydroxylase 4A14* (*Cyp4A14*) and *Metallothionein 2* (*Mt2*). We then prepared cDNA libraries from either control mouse livers or livers from mice that had been treated with 300 mg/kg APAP and harvested at 6hpe. Next, we used the miniPCR and GiS Viewer to determine expression of *Cyp4a14* (**Fig 3B**) and *Mt2* (**Fig 3C**) in control vs. 6hpe liver tissue. Quantification of *Cyp4a14* and *Mt2* RT-PCR reactions visualized in the GiS Viewer at cycles 21 and 29 respectively demonstrated that we were able to detect a significant upregulation of both these genes at 6hpe when compared to expression in control livers (**Fig 3E and 3F**), matching results from the single cell RNA-sequencing data.

To demonstrate the value of the GiS Viewer for use in low resource and remote environments, we next assessed its performance onboard the ISS. To validate that the heating and fluorescent imaging capabilities of the GiS Viewer were operational in space and not damaged during transport to the ISS, we employed reactions containing double-stranded DNA (dsDNA) oligos and EvaGreen® fluorescent double-stranded specific nucleic acid dye. When in the double-stranded state at room temperature, the dsDNA oligos are bound by EvaGreen, resulting in green fluorescence. Heating the reactions to 72˚C in the GiS Viewer causes denaturation of the dsDNA oligos and loss of green fluorescence, which is regained when the reactions are then cooled to room temperature. This test demonstrated that the GiS Viewer operated successfully on station (**S1 Video**).

Finally, we asked whether the GiS Viewer could be used in combination with the miniPCR thermal cycler to provide rapid readout of gene expression studies on-station without the need to return samples to Earth. To do so, we prepared matched *Cyp4a14* RT-PCR reactions to be run on the ground and in space. Ground controls were prepared at the same time as space reactions, using the same reagents. Samples launched to the ISS aboard the NG-16 mission and the reactions were run by astronaut Megan McArthur within one month of launch. The ground control reactions were run in parallel to the space reactions using the same model miniPCR as on the ISS and imaged in the GiS Viewer using both the same model iPad and the same imaging settings. The reactions were imaged at cycles 0, 21, and 35 (**Fig 4A**). An additional set of reactions was run to cycle 21 and then returned to Earth and analyzed via gel electrophoresis (**Fig 4A**).

When imaged at cycle 21, we were able to detect a statistically significant upregulation of *Cyp4a14* at 6hpe compared to control livers using the GiS Viewer and miniPCR thermal cycler on the ISS (**Fig 4B**), replicating the ground results. This change was also visible by eye. Interestingly, the 6hpe reactions run in space displayed dimmer fluorescence than the ground controls at both cycle 21 and cycle 35 (**Fig 4B**). To ask whether the observed differences in fluorescence were due to alterations in the efficacy of DNA amplification in space, we quantified gel band intensity on the reactions that had been run to cycle 21 on the ISS and then returned to Earth for gel electrophoresis. We observed no difference in intensity between space and ground samples in either the control or 6hpe reactions (**Fig 4C**), indicating that DNA amplification was not affected by the space environment.

## Discussion

This study introduces the GiS Viewer, a low-cost, portable tool for performing and viewing fluorescence-based biological assays. On Earth, we demonstrate that the GiS Viewer is an effective method to visualize results from RT-PCR and RT-LAMP assays and is able to visualize

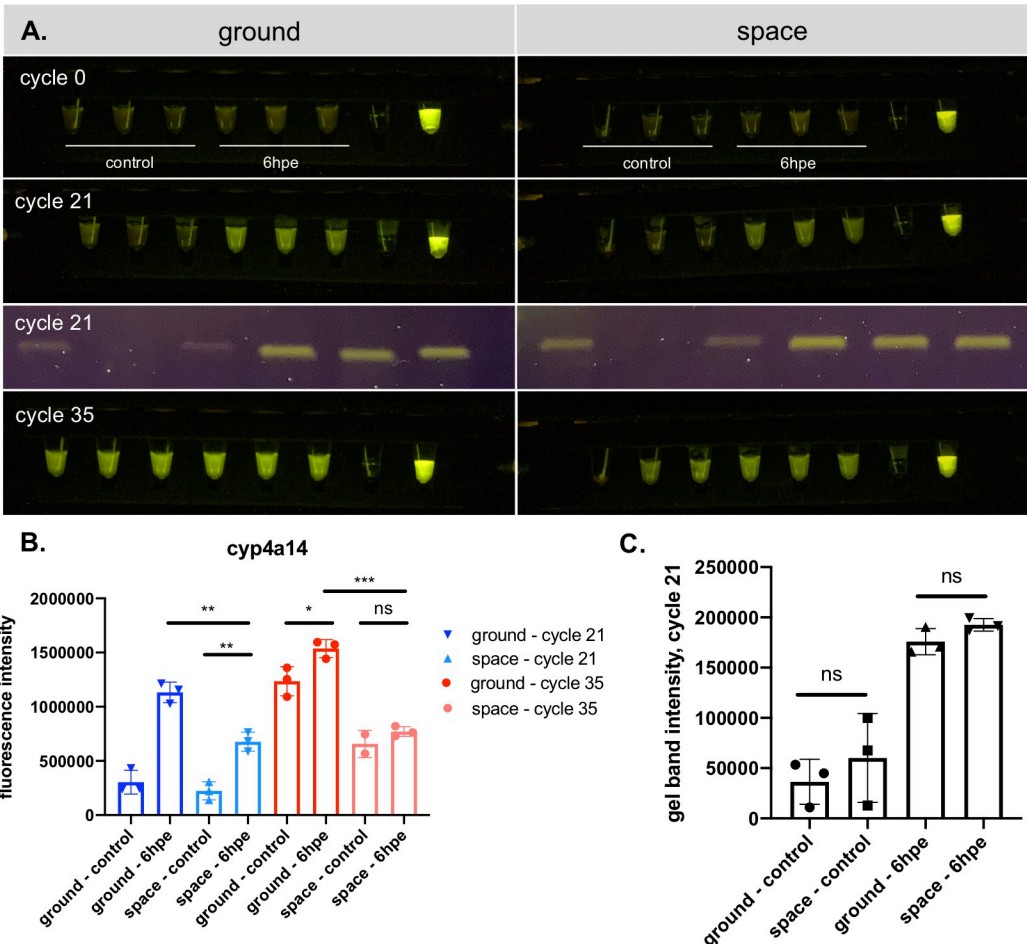

**Fig 4. GiS Viewer enables real-time, on-station analysis of drug-induced changes in gene expression. A.** *Cyp4a14* RT-PCR reaction run in the miniPCR and imaged in the GiS Viewer at indicated cycles. An additional reaction was run to cycle 21 and returned to earth for gel analysis (third panel). Left: ground controls, right: space samples. In each fluorescence image: samples 1–3: control liver cDNA from three separate mice, samples 4–6: 6hpe APAP liver cDNA from three separate mice, sample 5: water control, sample 6: 10 µM fluorescein. In each gel image: samples 1–3: control liver cDNA from three separate mice, samples 4–6: 6hpe APAP liver cDNA from three separate mice. All samples were run on the same gel and the gel image was spliced to reorder samples such that ground samples are on the left and space samples are on the right. **B.** Quantification of *Cyp4a14* fluorescence images at cycles 21 (blue) and 35 (red). **C.** Quantification of *Cyp4a14* gel at cycle 21. APAP: acetaminophen. hpe: hours post-exposure. Error bars are mean + SD. *$p < 0.05$, **$p < 0.01$, ***$p < 0.001$, ****$p < 0.0001$.

both 3–4 fold changes in gene expression and very low RNA copy number. Additionally, we show that the GiS Viewer is able to withstand the stresses of launch to the ISS and that it is a relevant tool for use in space, as samples could be analyzed in-flight without the need to return them to Earth. Together, these assays illustrate the power of the GiS Viewer as a rapid diagnostic and analytic tool for the ISS and terrestrial environments.

Though both the terrestrial and ISS trials using the GiS Viewer to analyze *Cyp4a14* RT-PCR reactions revealed the expected higher expression in APAP-treated mice, overall fluorescence levels were dimmer in space. It is unlikely that this was caused by damage to the GiS Viewer during launch, as its heating and imaging capabilities were verified via heating dsDNA oligos on the ISS prior to visualizing the liver cDNA. Significant reductions in amplification efficiency in space are also unlikely: gel electrophoresis was performed on the same samples once

returned to Earth, and no differences in band intensity relative to the Earth samples were observed. Because samples were imaged with the same model GiS Viewer and same model iPad on Earth and the ISS, these results suggest that procedural variability during image capture may have contributed to reduced fluorescence intensity in space. It is possible that slight variations in how the operator positioned the iPad in the adapter or in how the adapter mated to the GiS Viewer might explain these differences. In the future, integrating the GiS Viewer and imaging adapter into a single instrument may help reduce inter-run variability in detected fluorescence.

This study also highlights other areas for future optimization. While we were successful in performing critical steps of an RT-PCR workflow on the ISS, several steps were performed asynchronously on Earth, including DNA extraction and assembling RT-PCR reactions that were then frozen for launch. Additionally, although the GiS Viewer can visualize both red and green fluorescence, signal detection occurs on a single channel, meaning that for samples with both red and green fluorophores, the imaged fluorescence signal will be a combination of the two.

The current evaluation of the GiS Viewer for use in gene expression studies is timely: the NASA Twins Study revealed that a year in space caused thousands of disruptions to an astronaut's transcriptome [14]. The plethora of studies since–correlating spaceflight expression patterns with muscle mass, ocular degeneration, oxidative stress and more–have confirmed the importance of establishing methods for in-flight gene expression studies that are not subject to the delays and experimental variability induced by returning samples to Earth for analysis.

Terrestrially, the GiS Viewer finds a niche in point-of-care diagnostics, especially in regions lacking centralized lab testing facilities, where rapid field assessments may accelerate responses to public health crises. Similarly, new approaches in fluorescent environmental monitoring afford the chance for real-time evaluations of water for the presence of heavy metal ions and a variety of organic pollutants via molecular detection approaches. In sum, the GiS Viewer provides a rapid, sensitive, and user-friendly means to detect fluorescence readouts of multiple assay types.

## Materials and methods

### GiS Viewer

The GiS Viewer is a spaceflight-ready adaptation of the commercially available P51Dx device by miniPCR bio. The GiS Viewer allows visualization of fluorescence in the green and red spectra. The core of the instrument is an aluminum block that can receive a strip of eight or individual 0.2 mL tubes through openings on the top. Adhered to the back of the aluminum block is a thin-film polyimide heater. Temperature regulation is achieved through feedback from a thermistor located in proximity to the heater and tubes remain closed throughout the incubation. The GiS Viewer can be set to heat samples to either 72˚C or 65˚C. The side of the block facing the user has openings so that a portion of the tubes can be directly visualized and imaged while samples are incubated. The block is powder-coated with black paint to reduce reflection from the illumination source. A blue light source is positioned facing the openings on the block to excite the fluorophores in the samples, with a spectral excitation peak at 470 nm. There is no excitation filter. The emission filter is a 585 nm longpass filter. A control board is used to turn heating and illumination on and off, and to select one of two temperatures to which the block can be set. An enclosure made of FR4 fiberglass and acrylic holds all the elements in place. In addition, the instrument has a top cover to prevent light from entering the unit through the openings for the tubes, and a front window housing the emission filter through which the samples are observed. The maximum power used by the GiS Viewer is 10 watts. Its low power consumption makes the unit compatible with small USB-C chargers and

portable batteries. We estimate the total manufacturing cost of the GiS Viewer at production scale to be approximately $500 for parts and labor (data not shown).

## Animals

Three-month-old, male, C57BL/6J mice, purchased from Jackson Laboratories (Bar Harbor, ME, USA), were used in acute liver injury studies (APAP). All animals were housed in Association for Assessment and Accreditation of Laboratory Animal Care accredited facilities under a standard 12-hour light/dark cycle at 71˚F and 30–70% humidity with access to chow and water *ad libitum*. Studies were approved by the Brigham and Women's Hospital IACUC (Protocol Number 2016N000585).

## Acetaminophen (APAP) exposure

Mice were fasted 12 hours before administration of APAP. APAP was dissolved in warm 0.9% saline, and mice were injected with 300 mg/kg APAP, i.p. Food was returned to the mice after APAP treatment. Mice were euthanized at 6 hours post injection and then used for tissue harvest for further downstream analysis.

## Tissue harvest

Untreated (n = 3) and 6hpe APAP-treated mice (n = 3) were euthanized by cervical dislocation following carbon dioxide exposure. Liver tissue was harvested using standard surgical technique and flash frozen using liquid nitrogen.

## RNA extraction and RT-PCR

RNA was isolated from flash frozen murine livers via Trizol/Chloroform extraction. Prior to cDNA library preparation, RNA was DNase-digested with Invitrogen TURBO DNase™ (AM1907). 1ug RNA was used for each cDNA reaction and cDNA was prepared using the Bio-Rad iScript™ cDNA Synthesis Kit (1708891).

## RT-PCR using the miniPCR and GiS Viewer

RT-PCR reactions were carried out using the miniPCR 2x qGRN Master Mix (miniPCR bio). Reactions were run in the miniPCR (miniPCR bio), and samples were imaged in the GiS Viewer every 2 cycles between cycles 10–35. Cycling conditions were as follows: initial denaturation at 94˚C for 60 s then 18–35 cycles of 94˚C for 8 seconds, 55˚C for 8 seconds, 72˚C for 8 seconds. The miniPCR has a ramp rate of 2.4˚C/s for heating and 1.7˚C/s for cooling. Cooling is facilitated by forced air movement generated by fans. A thermistor attached to the heating element is used to measure the temperature, and a PID controller is used to modulate it. The maximum power used by the miniPCR is 80 watts. Samples were imaged at the extension step of the reaction and were held at 72˚C in the GiS Viewer during imaging. It is possible to image the reactions at room temperature if done within one minute of removing from the miniPCR, otherwise increasing background fluorescence that occurs as the reactions remain at room temperature can obscure the real signal. For each gene assayed, we selected the cycle number at which fluorescence intensity had not yet plateaued in any tube for downstream data analysis. No housekeeping control was used but input RNA and cDNA amounts were standardized across samples. Specifically, 1 μl of cDNA was used per reaction. For the 2-fold dilution experiment the following amounts of cDNA were used per reaction: 1 μl, 0.5 μl, 0.25 μl, 0.125 μl, 0.0625 μl, 0.03125 μl.

**Table 1. RT-PCR primer sequences.**

|  | F primer | R primer | Ref |
|---|---|---|---|
| *cyp4a14* | TGTTGCCATCTGGTCCCTAC | CACAACCAGCTCAGGAGTCAA | This study |
| *mt2* | CTATAAAGGTCGCGCTCCGC | GAGCAGGATCCATCGGAGG | This study |
| *18S* | GTAACCCGTTGAACCCCATT | CCATCCAATCGGTAGTAGCG | [15] |

## RT-PCR image analysis

Images were acquired on a 10.5″ iPad Pro 2nd Gen (MPDY2LL/A) using the Yamera imaging app with the following settings: focus 0.1, iso 700, shutter 1/40, tint -150, temp 4179. Fluorescence quantification was performed in ImageJ. Background fluorescence (water control or the area of the gel directly below the band for RT-PCR reactions vs. gels respectively) was subtracted prior to statistical analysis using Welch's t test to compare conditions. Data are graphed as mean with standard deviation. Primer sequences can be found in **Table 1**.

## SARS-CoV-2 LAMP

Synthetic RNA corresponding to SARS-CoV-2 Genebank sequence MN908947 (Twist Biosciences Cat # 102024) was used as a template in all reactions. Based on the manufacturer's reported concentration, samples containing 100, 500 and 1000 RNA molecules were prepared by serial dilutions from the stock solution. Reactions containing water served as no template controls (NTC) to monitor for any background signal. RT-LAMP reactions were performed using the WarmStart® LAMP Kit (New England Biolabs Cat # E1700) following the manufacturer's instructions without addition of the fluorescent dye. A set of primers targeting the N2 regions of SARS-CoV-2 together with a FAM-labeled molecular beacon [8] were used. The final concentration of the primer-beacon mix consisted of 1.6 μM forward inner primer (FIP), 1.6 μM backward inner primer f (BIP), 0.4 μM loop forward (LF), 0.4 μM loop backward (LB), 0.2 μM backward outer primer (B3), 0.2 μM forward outer primer (F3), and 0.25 μM molecular beacon. Primer sequences are described in [8]. Reactions with a total volume of 25 μl were prepared with the reagents described above and incubated at 65°C and visualized in the GiS viewer. Pictures were captured using a mobile phone.

## AT oligo reactions

The AT oligo reaction buffer comprised 0.1 M Tris pH 9.5, 3.75 μM EvaGreen®, and AT rich oligos at 3 concentrations (1.5 μM, 0.375 μM, 0.09375 μM, miniPCR bio). For imaging, the oligos were heated to 72°C for 10 min and then allowed to cool to room temperature for 10 min.

## Supporting information

**S1 Video. Demonstration of the heating capacity of the GiS Viewer onboard the ISS.** AT oligos (1.5 μM, 0.375 μM, 0.09375 μM) were heated to 72C for 10 minutes and then allowed to cool to room temperature for 10 minutes. Video speed is 100X real time. From left to right samples are two replicates of a 1:4 dilution series of AT oligos (1.5 μM, 0.375 μM, 0.09375 μM) followed by a water control and 10 μM fluorescein.
(MP4)

**S1 Raw image. Raw image for gel presented in Fig 4A.** Sample order: 1–3. Space control. 4–6: Space 6hpe. 7–9: Ground control. 10–12: Ground 6hpe.
(PDF)

**S2 Raw image. Raw image for gel presented in Fig 4A, without labeling.** Sample order: 1–3. Space control. 4–6: Space 6hpe. 7–9: Ground control. 10–12: Ground 6hpe.
(TIFF)

**S1 Data. Raw data points for the fluorescence intensity graphs displayed in Fig 3, panels D-F, and Fig 4, panels B, C.**
(XLSX)

## Acknowledgments

We thank Jessica Quenzer from Stuyvesant High School in New York, NY for her advice and support and Astronaut Megan McArthur for completing this experiment aboard the ISS. This work was supported in part by New England Biolabs and the ISS U.S. National Laboratory.

## Author Contributions

**Conceptualization:** Kristoff Misquitta, Bess M. Miller, Kathryn Malecek, Emily Gleason.

**Formal analysis:** Bess M. Miller.

**Investigation:** Bess M. Miller, Emily Gleason.

**Methodology:** Bess M. Miller, Emily Gleason, Chad M. Walesky, Ezequiel Alvarez Saavedra.

**Project administration:** Emily Gleason, Kathryn Martin, Kevin Foley, D. Scott Copeland, Ezequiel Alvarez Saavedra, Sebastian Kraves.

**Supervision:** Kathryn Malecek, Emily Gleason, Ezequiel Alvarez Saavedra, Sebastian Kraves.

**Validation:** Emily Gleason.

**Writing – original draft:** Kristoff Misquitta, Bess M. Miller.

**Writing – review & editing:** Bess M. Miller, Kathryn Martin, Ezequiel Alvarez Saavedra, Sebastian Kraves.

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
