## [Decision Letter · Decision Letter 0]

10 Oct 2023

PONE-D-23-27008A fluorescence viewer for rapid molecular assay readout in space and low-resource terrestrial environmentsPLOS ONE

Dear Dr. Kraves,

Thank you for submitting your manuscript to PLOS ONE. After careful consideration, we feel that it has merit but does not fully meet PLOS ONE’s publication criteria as it currently stands. Therefore, we invite you to submit a revised version of the manuscript that addresses the points raised during the review process.

We look forward to receiving your revised manuscript.

Kind regards,

Basant Giri, Ph.D.

Academic Editor

PLOS ONE

Journal Requirements:

We thank Jessica Quenzer from Stuyvesant High School in New York, NY for her advice and support and Astronaut Megan McArthur for completing this experiment aboard the ISS. This work was supported in part by New England Biolabs and the ISS U.S. National Laboratory.

Reviewers' comments:

Reviewer's Responses to Questions

**Comments to the Author**

1. Is the manuscript technically sound, and do the data support the conclusions?

Reviewer #1: Yes

Reviewer #2: Yes

2. Has the statistical analysis been performed appropriately and rigorously? 

Reviewer #1: Yes

Reviewer #2: Yes

3. Have the authors made all data underlying the findings in their manuscript fully available?

Reviewer #1: Yes

Reviewer #2: Yes

4. Is the manuscript presented in an intelligible fashion and written in standard English?

Reviewer #1: Yes

Reviewer #2: Yes

5. Review Comments to the Author

Reviewer #1: In this study, the authors presented the Genes in Space Fluorescence Viewer (also called GiS Viewer) which can be used to visualize fluorescence in the red and green ranges. They have demonstrated this viewer as a portable assay readout and can be used to visualize fluorescence in space.

The work is exciting and well-written, and most parts are well-explained to the reader. I have minor comments which, upon addressed, could probably help to improve the quality of the paper. I would expect following minor comments to be addressed before the work can be published.

1. Abstract (and methods, Figure 2): Author mentioned that heater is inside the PCR tube. Also, it seems so from the Figure 2. If that’s the case, how did you manage to control same temperature in all tubes at a same time? Moreover, what about risk of contamination? Reusability issue can be another factor. Please clarify.

2. Any specific Excitation/ emission wavelength was used?

3. In conclusion the authors mentioned about low power consumption. Please specify the power used for a PCR cycle or for the whole process.

4. How did you manage to control of specific temperature of PI heater?

5. What is the ramp rate during heating/cooling process?

6. What element was used for cooling during PCR cycle?

Reviewer #2: I find the research work "A fluorescence viewer for rapid molecular assay readout in space and low-resource terrestrial environments" important as it provides a low-cost alternative to visualize multiple assay types in fluorescence mode run in standard PCR tubes. However, following comments should be addressed before it can be considered for publication.

1) Analytical performance of the proposed method/device is not reported. It is important to clearly mention the basic analytical parameters such as LOD, sensitivity and measurement time. If possible, compare these parameters with the other low-cost florescence readout systems. Also provide a tentative cost estimate of the device.

2) Mention study limitations of the method in an appropriate place.

3)Figure 1A is difficult to follow. Please improve the figure quality; For example, a 3D schematic of the setup would help.

4) Missing experimental details (Lines 14-25 page 10): Provide detail information of the excitation and collection filters, light source and heater used in the setup. Figure 1 is not cited in the lines 14-25 in page 10.

5) Introduction section needs improvements. With more references cited, make it easy to follow. For examples, several references can be cited in the first paragraph of introduction section.

6. PLOS authors have the option to publish the peer review history of their article (what does this mean?). If published, this will include your full peer review and any attached files.

Reviewer #1: No

Reviewer #2: No

---

## [Author Response · Author response to Decision Letter 0]

21 Nov 2023

REVIEWER #1

1. Abstract (and methods, Figure 2): Author mentioned that heater is inside the PCR tube. Also, it seems so from the Figure 2. If that’s the case, how did you manage to control same temperature in all tubes at a same time? Moreover, what about risk of contamination? Reusability issue can be another factor. Please clarify.

We thank the reviewer for highlighting several areas in the description of the GiS Viewer that could benefit from additional detail. We have updated the methods section to clarify the placement of the heating element within the GiS Viewer. The PCR tubes are placed within an aluminum block and a thin-film polyimide heater is adhered to the back of the aluminum block. Temperature regulation is then achieved through feedback from a thermistor located in proximity to the heater. The tubes remain closed throughout the incubation.

2. Any specific Excitation/emission wavelength was used?

We have updated the methods section to clarify that fluorophores are excited by blue LEDs with a spectral peak at 470 nm. 

3. In conclusion the authors mentioned about low power consumption. Please specify the power used for a PCR cycle or for the whole process.

We have updated the methods to include maximum power data for both the GiS Viewer and the miniPCR.

4. How did you manage to control of specific temperature of PI heater?

We have updated the relevant methods sections to specify that temperature is controlled using a thermistor located in proximity to the heater in the GiS Viewer and a NTC thermistor with PID control in the miniPCR.

5. What is the ramp rate during heating/cooling process?

We have updated the methods to specify that the ramp rates in the miniPCR are 2.4°C/s for heating and 1.7°C/s for cooling.

6. What element was used for cooling during PCR cycle?

We have updated the methods to specify that fans are used for cooling in the miniPCR. 

REVIEWER #2

1. Analytical performance of the proposed method/device is not reported. It is important to clearly mention the basic analytical parameters such as LOD, sensitivity and measurement time.

If possible, compare these parameters with the other low-cost florescence readout systems. Also provide a tentative cost estimate of the device.

We agree that these are important parameters. We have updated the text to specify the limit of detection for the RT-PCR and LAMP assays used here and the timeframes needed for detection. For the SARS-CoV-2 LAMP assay, we were able to detect 100 copies of viral RNA (lowest amount tested). For qPCR, we were able to detect a 3-4 fold difference in gene expression. Measurement time is assay dependent and we have updated the text to better clarify the imaging process. For assays that are continuously incubated at either room temperature, 72°C, or 65°C, imaging can be done in parallel with incubation and requires only turning on the LEDs and capturing an image using a phone or tablet. For assays that require incubation at varying or other temperatures, samples first have to be transferred into the GiS Viewer, which takes under a minute. We have also updated the text to specify the approximate manufacturing cost of the Viewer. 

2. Mention study limitations of the method in an appropriate place.

We have added a section in the discussion detailing the major limitations of the study and the GiS Viewer. 

3. Figure 1A is difficult to follow. Please improve the figure quality; For example, a 3D schematic of the setup would help.

We agree that Figure 1A would benefit from clarification. We have replaced the schematics with ones that better show the 3D structure of the GiS Viewer and the removable phone/tablet holder. 

4. Missing experimental details (Lines 14-25 page 10): Provide detail information of the excitation and collection filters, light source and heater used in the setup. Figure 1 is not cited in the lines 14-25 in page 10.

We thank the reviewer for pointing out additional details needed in the methods section. These have now been added and Figure 1 has been cited as well. 

5) Introduction section needs improvements. With more references cited, make it easy to follow. For examples, several references can be cited in the first paragraph of introduction section.

We have updated the introduction section to include additional citations to better situate the reader in the current context of the field.

---

## [Editor Report · Decision Letter 1]

8 Dec 2023

A fluorescence viewer for rapid molecular assay readout in space and low-resource terrestrial environments

PONE-D-23-27008R1

Dear Dr. Kraves,

We’re pleased to inform you that your manuscript has been judged scientifically suitable for publication and will be formally accepted for publication once it meets all outstanding technical requirements.

Kind regards,

Basant Giri, Ph.D.

Academic Editor

PLOS ONE
---

## [Editor Report · Acceptance letter]

23 Jan 2024

PONE-D-23-27008R1 

PLOS ONE

Dear Dr. Kraves, 

I'm pleased to inform you that your manuscript has been deemed suitable for publication in PLOS ONE. Congratulations! Your manuscript is now being handed over to our production team.

Kind regards, 

on behalf of

Dr. Basant Giri 

Academic Editor

PLOS ONE